# Anomalous Strain Recovery after Stress Removal of Graded Rubber

**DOI:** 10.3390/polym14245477

**Published:** 2022-12-14

**Authors:** Quoc-Viet Do, Takumitsu Kida, Masayuki Yamaguchi, Kensuke Washizu, Takayuki Nagase, Toshio Tada

**Affiliations:** 1School of Materials Science, Japan Advanced Institute of Science and Technology, 1-1 Asahidai, Ishikawa, Nomi 923-1292, Japan; 2Material Research & Development HQ, Sumitomo Rubber Industries, Ltd., 1-1, 2-Chome, Tsutsui, Chuo, Hyogo, Kobe 651-0071, Japan

**Keywords:** graded rubber, rubber elasticity, styrene–butadiene rubber

## Abstract

Mechanical responses after the uniaxial deformation of graded styrene–butadiene rubber (SBR) with a gradient in the crosslink points in the thickness direction were investigated as compared with those of homogenously vulcanized SBR samples. The elongational residual strain of a graded sample was found to depend on the part with a high crosslink density. Therefore, it showed good rubber elasticity. After stress removal, moreover, the graded sample showed a marked warpage. This suggested that shrinking stress acted on the surface with a high crosslink density, which would avoid a crack growth on the surface. The sample shape was then recovered to be flat very slowly, indicating that the shrinking stress worked for a long time. This unique rubber elasticity, i.e., slow strain recovery with an excellent strain recovery, makes graded rubber highly significant.

## 1. Introduction

A rubber has a crosslinked structure, in which crosslink points are homogeneously distributed in general. Strictly speaking, however, most rubbers have a gradient in the crosslink density, especially in a thick product, because each part in a rubber product has different thermal histories. Since polymeric materials including rubbers usually have a low thermal diffusivity, it takes a long time to be in an equilibrium temperature profile at vulcanization process [1]. Therefore, a core region in a rubber product must have a short exposure period at a high temperature compared with a skin region. This may result in the difference in the crosslink density, although its effect on the mechanical properties can be ignored for most rubber products used in industry. Such a situation is, however, pronounced and should be considered when vulcanization occurs slowly. Bellander et al. [2] vulcanized styrene–butadiene rubber (SBR) without crosslinking agents such as sulfur and found a gradient in the crosslink density. A rubber with a crosslink gradient, i.e., a graded rubber, may show poor rubber elasticity when the stress is applied in the gradient direction. This is reasonable because a layered part with no/few crosslink points will flow by applied normal force. From the viewpoint of energy absorption, however, a number of researchers reported that damping properties were improved by providing a crosslink gradient [3,4,5,6]. This must be attributed to dangling chains in the region with a low crosslink density, which showed prolonged relaxation modes with a high level of energy absorption [7,8,9,10].

A graded rubber can be obtained by different methods, including the lamination of different layers [11] and manipulated photo-curing [12,13,14]. For conventional rubber materials, Ikeda prepared a graded SBR by vulcanizing laminated sheets with different sulfur contents [15,16]. She found that the mechanical properties such as tensile properties were mostly decided by the area with a high crosslink density. Moreover, Glebova et al. [17] revealed that the crosslink density in SBR around zinc oxide particles was high in the nanoscale regions (ca. 200 nm). Considering that nanoparticles can show interphase transfer between different rubbers [18], this technique may provide a new idea to make a graded rubber in the future. Finally, Wang et al. [19] proposed a simple method to prepare a graded SBR by diffusing sulfur from one surface, followed by vulcanization. Since diffusion constants of curatives have been studied for a long time after the pioneering works by van Amerongen [20] and Gardiner [21], this technique must be noted. According to Wang et al., the obtained graded SBR showed high values of loss tangent in a wide temperature range due to broad distribution of glass transition temperature *T_g_* [19]. Besides the pronounced energy absorption, however, attractive properties of a graded rubber composed of conventional materials have not been reported yet to the best of our knowledge.

Here, we prepared a graded SBR by vulcanizing under a temperature gradient and found unique strain recovery behaviors after stress removal. Dynamic mechanical properties, as well as tensile properties including stress relaxation behaviors, were also investigated in detail.

## 2. Materials and Methods

### 2.1. Materials

Styrene–butadiene rubber (SBR) with a styrene content of 25 wt.% was kindly supplied from Sumitomo Rubber Industries Ltd. (Kobe, Japan). The weight-average molecular weight as polystyrene standard was 250,000. Furthermore, carbon blacks (Diablack-H; Mitsubishi Chemical, Tokyo, Japan), stearic acid (NOF Corporation, Tokyo, Japan), zinc oxide (Fujifilm Wako Pure Chemical Corporation, Osaka, Japan), *N*-phenyl-*N*’-(1,3-dimethylbutyl)-*p*-phenylene diamine as an antioxidant (Nocrac 6C; Ouchi Shinko Chemical Industrial, Tokyo, Japan), aroma oil (VivaTec 500; H&R, Hamburg, Germany), sulfur (Tsurumi Chemical Industry, Tokyo, Japan), *N*-cyclohexyl-2-benzothiazolyl sulfenamide (CBS, Nocceler CZ-G; Ouchi Shinko Chemical Industrial), and 1,3-diphenylguanidine (DPG, Nocceler D; Ouchi Shinko Chemical Industrial) were employed. All of them were used without further purification.

### 2.2. Sample Preparation

The sample recipe is shown in Table 1. Mixing was performed by three steps. At the first step, all ingredients except for sulfur and accelerators, such as CBS and DPG, were added into a 1700 cc internal mixer (Mixtoron BB; Kobelco, Kobe, Japan) and mixed at 77 rpm for 3 min. The initial temperature of the mixer was controlled at 30 °C, and the final temperature was about 150 °C. After mixing, the mixture was taken out and cooled down at room temperature. Then, it was put into the mixer at 30 °C again with sulfur and accelerators and mixed at 44 rpm for 3 min as the second step. The final temperature was about 100 °C. Finally, the obtained mixture was kneaded by an 8-inch two-roll mill (Kansai Roll, Osaka, Japan) at 60 °C to prepare a sheet with a 2.5 mm thickness.

Sample sheets were exposed to high temperatures in a compression molding machine under 10–20 MPa for 10 min. The temperature conditions with sample codes are shown in Table 2. In the sample codes, “H” represents the homogeneously crosslinked samples, i.e., both plates of the compression molding machine were controlled at the same temperature, whereas “G” denotes the graded samples. The thickness of the compressed sheets was about 1.3 mm. After the vulcanization process, the surface temperature was measured again, which is also summarized in Table 2. The sample code “UV” represents the unvulcanized sample, although it was compressed at 80 °C for 30 s under 20 MPa to reduce the thickness to 1.3 mm. All samples were kept at room temperature at least 24 h before testing.

### 2.3. Measurements

Vulcanization behaviors at various temperatures were evaluated by a rotorless curemeter (Curelastometer Type R 7; Eneos Trading Company, Tokyo, Japan) following ISO 6502.

The temperature dependence of tensile moduli was measured by a dynamic mechanical analyzer (Rheogel E4000; UBM, Muko, Japan). The frequency and heating rate were 10 Hz and 2 °C/min, respectively. The specimen had the following dimensions: 4 mm in width, 15 mm in length, and 1.3 mm in thickness.

Tensile tests were carried out by a tensile testing machine (Autograph AGS-X; Shimadzu, Kyoto, Japan) at 25 °C. The dumbbell-shaped specimens, No.7 of JIS-K6251 (corresponded to ISO37-4), were cut from the sheets using a dumbbell sample cutting machine (Super Dumbbell Cutter SDL-200; Dumbbell, Kawagoe, Japan). The crosshead speed was 100 mm/min, and the initial distance between two clamps was 20 mm. Three measurements were conducted for each sample.

Stress relaxation measurements were performed using the tensile machine. Similar to the tensile tests, the dumbbell-shaped specimens with an initial gauge length of 11 mm were stretched at 100 mm/min. The stretching was stopped at a strain of 1.0 and kept for 900 s to measure the stress relaxation. After the stress relaxation measurements, the samples were taken out from the tensile machine and kept at 25 °C to investigate the strain recovery behaviors. The sample shape was recorded by a digital camera (HDR-CX540V; Sony, Tokyo, Japan) to characterize the recovery process.

## 3. Results and Discussion

Figure 1 shows the torque curves versus curing time *t_c_* at various temperatures. The torque did not increase at all in 10 min at/below 110 °C, suggesting that no/little vulcanization reaction occurred at these temperatures. Beyond 135 °C, the torque increased with the curing time beyond 300 s. Furthermore, it was found that the optimum cure time *t*_90_ was around 140 s at 170 °C.

The temperature dependence of tensile storage modulus *E’* and loss tangent *tan δ* at 10 Hz is shown in Figure 2. The glassy, transition, and rubbery regions were clearly detected for all samples.

Although there was no/little difference in the *E’* values among the samples in the glassy region, the curves in the transition region were slightly different. As seen in Figure 2a, *E’* decreased due to the glass-to-rubber transition that occurred at a high temperature for H170, i.e., fully crosslinked sample. Figure 2d showed that the values of *tan δ* for H170 were lower than those for the others in the temperature range from −40 to −5 °C. As the vulcanization temperature increased, providing more crosslink points, segmental motion in the rubber was more restricted, which resulted in high *T_g_* [7,8,9,10,19]. The number of crosslink points, of course, affected the modulus in the rubbery region. In Figure 2b, *E’* values from 20 to 60 °C were plotted. The order of *E’* values corresponded with the vulcanization temperature for the homogeneously crosslinked samples. *E’* values of the graded rubbers, i.e., G170-80 and G170-110, were found to be between those of H110 and H170. Regarding *tan δ*, the graded samples did not show high values in this study, which was different from some reports [8,9,10,12,19]. Presumably, the present samples had few dangling chains compared with those used in the previous studies. Therefore, the energy absorption was not largely expected, at least in the linear viscoelastic range. Moreover, Figure 2 indicates that H80 had almost no crosslink points because the dynamic mechanical properties were similar to those of the unvulcanized one, i.e., UV. As a result, they showed high *tan δ* values in the high-temperature region (Figure 2c).

Stress–strain curves are shown in Figure 3. The stresses are engineering values, i.e., force divided by the initial cross-sectional area, while the strains are also engineering values, i.e., distance divided by the initial distance. The measurements were performed three times for each sample. Since the experimental error was not large for all samples, we showed the middle values of each in the figure. The stresses for the graded samples were between those of H170 and H110, suggesting that the part with a high crosslink density, which must be the surface region exposed to the high temperature at the vulcanization process, was responsible for the stress generation [15]. They were reasonable results and corresponded with Figure 2.

For some samples, stretching was stopped at a strain of 1.0 to measure the stress relaxation behaviors. Figure 4 shows the stress relaxation curves normalized by the stress at the cessation of stretching. The horizontal axis represents the time after the cessation of stretching. It was found that the graded samples showed high levels of normalized stress in the long-time region. The values were comparable with that of H170 and much better than that of H110, indicating that the graded samples showed good rubber elasticity. Similar to stress generation at stretching, the part with a high crosslink density was largely responsible for the rubber elasticity. It was also found from the figure that H80 showed higher values than UV, suggesting that a weak network existed in H80.

After 900 s, the samples were taken out from the tensile machine and kept at room temperature to evaluate the strain recovery property. The sample shapes immediately after stress removal are exemplified in Figure 5. As shown in Figure 5a, simple shrinkage with a flat shape was detected for H170, as expected. However, the graded samples exhibited marked bending deformation (Figure 5b). The inner side was the part with a high crosslink density. The bending deformation must be attributed to the difference in the recovery stress acting on each surface of the graded samples. After the stress removal, the side with a high crosslink density was exposed to a high shrinking stress, which could avoid a crack growth on the surface. In contrast, the opposite side had no or weak shrinking stress, owing to the orientation relaxation of chain segments.

Eventually, the graded samples became flat, as shown in Figure 6. However, the bending deformation was still detected even after one week for both graded samples. For a better understanding of the recovery process, the recovery ratio *R*(*t_r_*) was calculated using the following equation as a function of the time after stress removal *t_r_*:(1)$$R\left(t_{r}\right)\;\left(\%\right)=\left(1-\frac{\varepsilon \left(t_{r}\right)}{\varepsilon_{i}}\right)\times 100$$
where *ε*(*t_r_*) is the strain at time *t_r_* and *ε_i_* is the initial strain applied by stretching. In this experiment, *ε_i_* is 1. For bended samples, the outer and inner lengths were evaluated by Image J software using the pictures, and the average values were used as *ε*(*t_r_*).

Figure 7 shows the recovery curves. H170 showed an immediate recovery. This is reasonable because the experiments were performed well beyond *T_g_*. The value was 98 ± 1%, suggesting that the residual strain, 1−*R*(∞), was around 2%. As the crosslink density for the homogenously crosslinked rubbers decreased, the time dependence became obvious, and the equilibrium values became small, i.e., poor rubber elasticity. In other words, samples showing a small residual strain exhibited a quick recovery at room temperature. As compared with the homogenously crosslinked samples, the graded ones showed a different behavior. They exhibited good strain recovery at equilibrium conditions, i.e., 96 ± 1% for G170-80 and 97 ± 1% for G170-110, which was much better than that for H110 (92%). However, it took a long time to recover; i.e., the recovery ratio slowly increased with *t_r_*, especially for G170-80 with bending deformation. This suggests that the shrinking stress acting on the surface with a high crosslink density would work for a long time. This slow recovery cannot be predicted from the stress relaxation data because the relaxation curves were not much different from that of H170, as shown in Figure 4. In the graded rubber, segmental orientation in the side with a low crosslink density was mostly relaxed when the stress was removed. Therefore, during recovery, the shrinking stress was applied from the other side with dense crosslink points. As a result, the reorganization of segments was required in the weakly crosslinked side, which must be the origin of slow recovery.

## 4. Conclusions

In this study, graded rubbers were prepared by vulcanizing a conventional SBR under a temperature gradient. The obtained samples had a gradient in crosslink density in the thickness direction. Although the dynamic mechanical properties including *tan δ* were not much different from those of the homogeneously crosslinked samples, the rubber elasticity was found to be different. After the cessation of stretching, stress relaxation behavior was evaluated as compared with homogeneously crosslinked rubbers. It was found that the stress was hardly relaxed for the graded rubbers during stress relaxation measurements, which was comparable to the fully crosslinked rubber. This must be attributed to the part with a high crosslink density. Furthermore, after stress removal, the graded samples showed marked bending deformation due to the mismatch in the shrinking stress between both surfaces. Eventually, the samples showed less bending deformation and became flat, which took a long time, even well beyond *T_g_*. Moreover, they showed good strain recovery, i.e., low residual strain, even though shrinking occurred very slowly. The segmental motion in the part with a low crosslink density must be the origin of slow strain recovery. This must be a common phenomenon for graded rubbers with a crosslink gradient in the thickness direction.

## Figures and Tables

**Figure 1 polymers-14-05477-f001:**
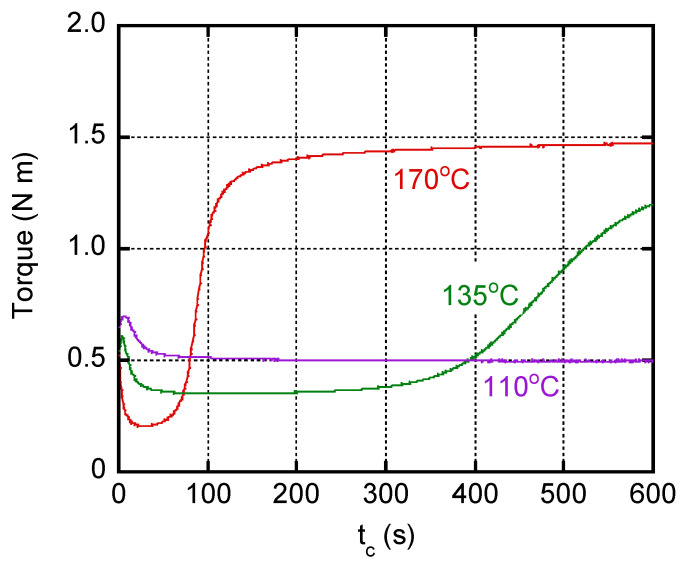
Torque curves plotted against curing time *t_c_* at various temperatures.

**Figure 2 polymers-14-05477-f002:**
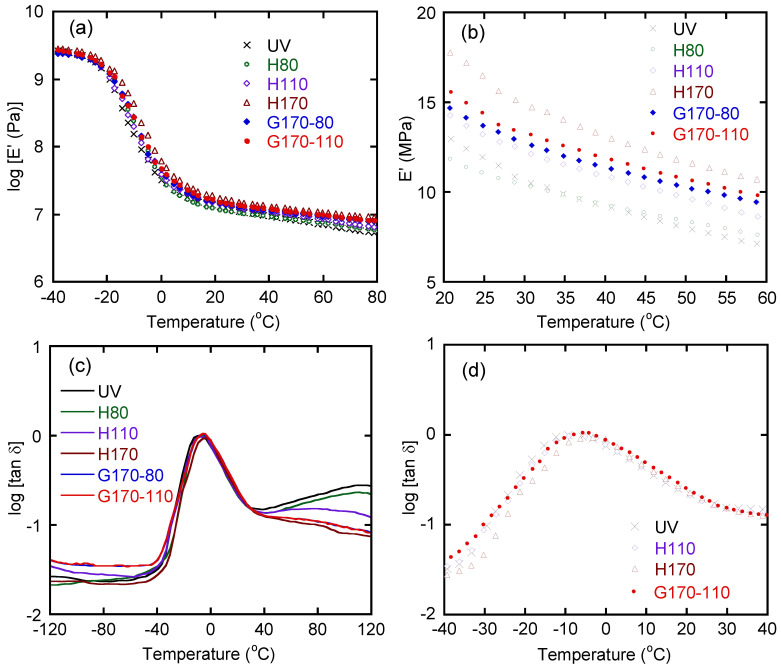
Temperature dependence of (**a**) and (**b**) tensile storage modulus *E’* and (**c**) and (**d**) loss tangent *tan δ* at 10 Hz.

**Figure 3 polymers-14-05477-f003:**
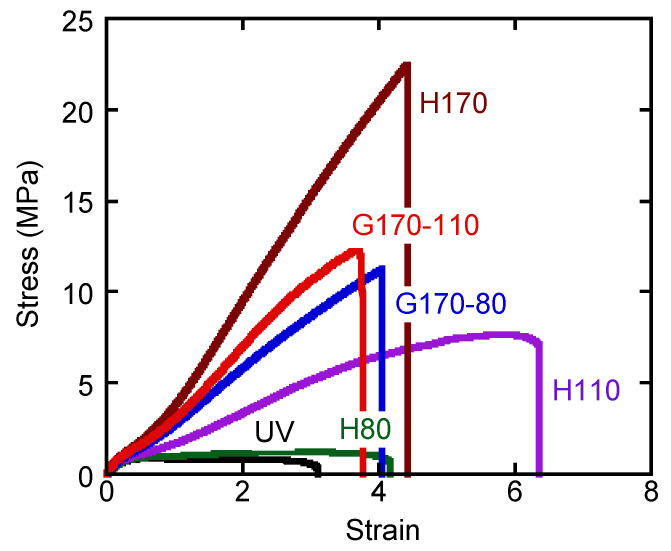
Stress–strain curves.

**Figure 4 polymers-14-05477-f004:**
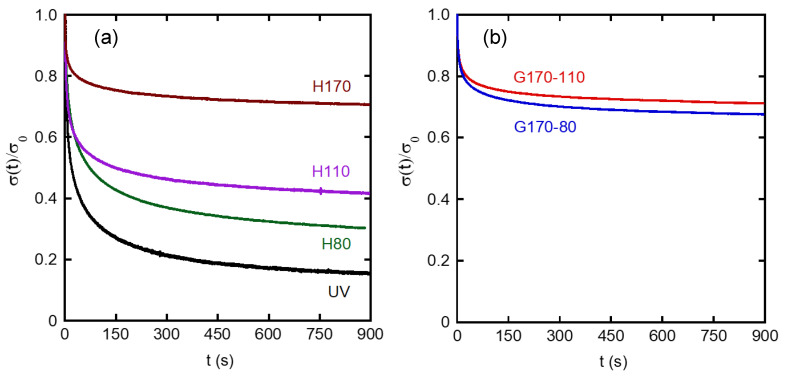
Relaxation curves of normalized stress, i.e., stress *σ*(*t*) divided by that at the cessation of stretching at *ε* = 1 *σ*_0_ for (**a**) homogenous samples and (**b**) graded samples.

**Figure 5 polymers-14-05477-f005:**
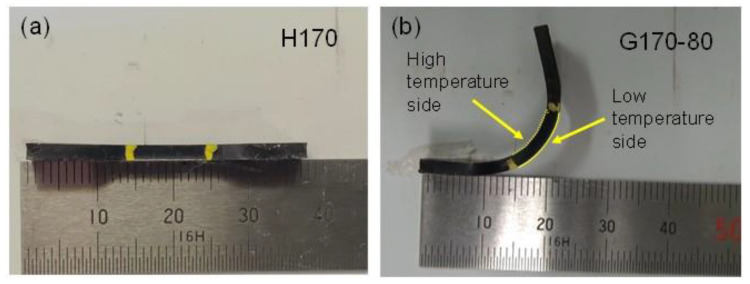
Sample shapes immediately after removal from the tensile machine for (**a**) H170 and (**b**) G170-80. The pictures were taken from a side view of the samples.

**Figure 6 polymers-14-05477-f006:**
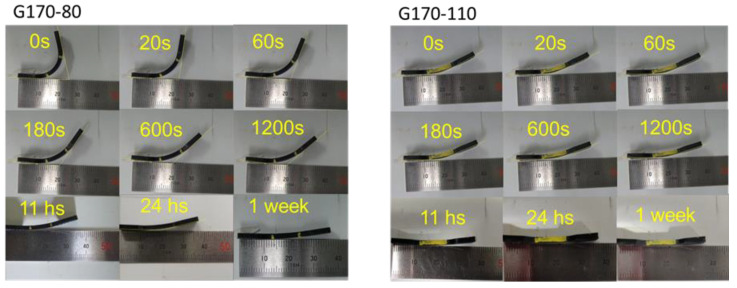
Recovery process after stress removal.

**Figure 7 polymers-14-05477-f007:**
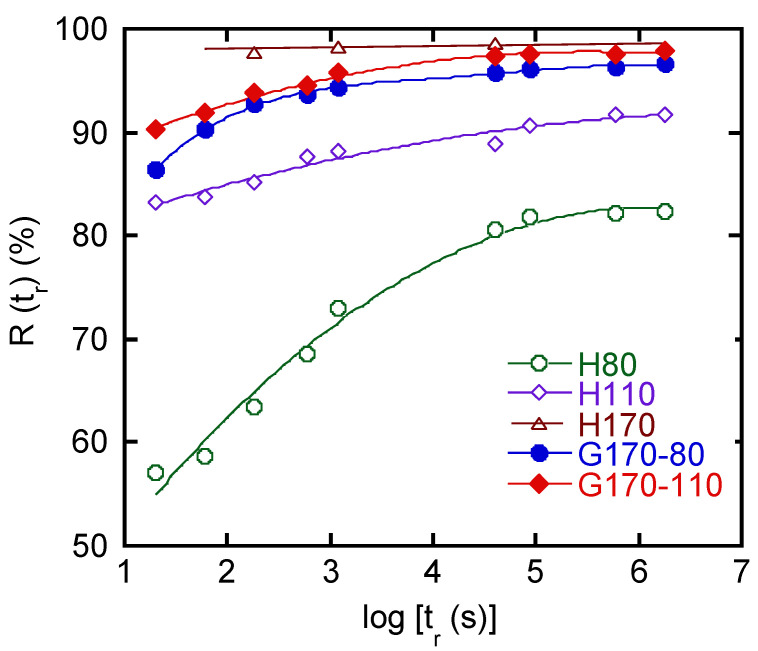
Recovery ratio *R*(*t_r_*) as a function of time after stress removal.

**Table 1 polymers-14-05477-t001:** Recipe of SBR compound.

Ingredients	Amount (phr)
SBR	100
Carbon black	50
Stearic acid	2
Zinc oxide	3
Antioxidant	1
Aroma oil	5
Sulfur	1.45
CBS *	2.3
DPG **	1.85

* *N*-Cyclohexyl-2-benzothiazole sulfenamide. ** Diphenyl guanidine.

**Table 2 polymers-14-05477-t002:** Sample codes and temperature conditions at compression molding.

Sample Codes	Set Temperature (°C)		Temperature at the End of Vulcanization Process (°C)	
	Top Plate	Bottom Plate	Top Plate	Bottom Plate
UV	-	-	-	-
H80	80	80	80	80
H110	110	110	110	110
H170	170	170	170	170
G170-80	170	50	170	80
G170-110	170	80	170	110

## Data Availability

Not available.

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
