# Peer review of "Anomalous Strain Recovery after Stress Removal of Graded Rubber"

_polymers, 2022, doi:10.3390/polym14245477_

Round 1

Reviewer 1 Report

The manuscript "Anomalous Strain Recovery after Stress Removal of Graded Rubber" by Quoc et al. reports the dynamic mechanical properties of graded rubber samples and tensile properties, including stress relaxation behaviors. It is unique because the attractive properties of a graded rubber composed of conventional materials have not been reported. The contents are informable and logical, but the referee suggests minor revision and English refinement before it can be published. There are some points that concern the reviewer as follows: 

L. 101: (Curelastometer; Eneos Trading Company, Tokyo, Japan)  => The type of Curelastometer should be shown. Curelastometer Type R 7?

L. 107: The dumbbell-shaped specimens, No.7 of JIS-K6251, => corresponded ISO 37-4 should be mentioned.

L. 123: the vulcanization period => Is this the optimum cure time T90?

L. 128: Figure 2 (a) => Not clear marks.

L. 141: tan d => tan delta (font error)

L. 141: Regarding tan d, the graded samples did not show high values in this study. => The referee does not understand why the authors regard the "tan delta" did not show high values in Fig. 2 (c) or (d).

L. 148: Figure 3: The sample number N the authors tested should be mentioned.

L. 151: nominal values => The referee needs help understanding this term. The meaning of "nominal values" the authors intended should be explained. 

L. 177: Figure 5 => The shoot direction of photos should be mentioned. Sample's cross-sectional direction?

L. 183: unnecessary indent

L. 212: tan d => tan delta (font error)

L. 216: found that the "stress" was hardly => found that the "strain" was hardly? 

References

Ref [2]: Bellander, M.; Stenberg, B.; Persson, S.J.P.E. Crosslinking of polybutadiene rubber without any vulcanization agent. Science 1998, 237 38, 1254-1260. 

=> Bellander, M.; Stenberg, B.; Persson, S. Crosslinking of polybutadiene rubber without any vulcanization agent. Polym. Eng. Sci. 1998, 237 38, 1254-1260.

The referee's other questions:

1. The authors focused on crosslinking conditions. Why did the authors perform some swelling tests?

2. In Figure 3, it seems strange that the strain for UV and H80 samples are so small compared with other vulcanized samples. Why? The referee believes that typically, they showed more considerable strain, ex.  8 or 10. 

3. As for Figure 3, usually, it is hard to cut unvulcanized samples. Is it possible to explain the procedure of sample preparation, such as UV and H80 samples?

Reviewer 2 Report

In this manuscript the authors have shown the strain recovery behavior of a graded styrene butadiene rubber containing a gradient cross-linking density in the direction perpendicular to the tensile direction. It will be an interesting paper. The methodology is clear and the conclusion is well supported by experimental results. However, in my opinion, some expressions should be revised more clearly and some data should be re-check, as followings:

1.       Lines 215, 216,217:

It was found that the stress was hardly relaxed for the graded rubbers, which was comparable with the fully crosslinked rubber. This must be attributed to the part with a high crosslink density.”

In fact, it can be said that the root cause of the phenomenon mentioned in the article is that the surface layer with fewer crosslinks shinks more slowly, leading to the difference in the recovery speed of the two sides and the behavior of dangling chains at transition layer of the grade samples.

2.       The stress-strain curve of H110 sample shown in Figure 3 (line 148) seem to be not consistent with the torque curve at 110 oC shown in Figure 1 (line 118).
